# Posttranslational chemical installation of azoles into translated peptides

Haruka Tsutsumi[1,2], Tomohiro Kuroda[1,2], Hiroyuki Kimura [1], Yuki Goto [1✉] & Hiroaki Suga [1✉]

Azoles are five-membered heterocycles often found in the backbones of peptidic natural products and synthetic peptidomimetics. Here, we report a method of ribosomal synthesis of azole-containing peptides involving specific ribosomal incorporation of a bromovinylglycine derivative into the nascent peptide chain and its chemoselective conversion to a unique azole structure. The chemoselective conversion was achieved by posttranslational dehydro-bromination of the bromovinyl group and isomerization in aqueous media under fairly mild conditions. This method enables us to install exotic azole groups, oxazole and thiazole, at designated positions in the peptide chain with both linear and macrocyclic scaffolds and thereby expand the repertoire of building blocks in the mRNA-templated synthesis of designer peptides.

[1] Department of Chemistry, Graduate School of Science, The University of Tokyo, Bunkyo, Tokyo, Japan. [2]These authors contributed equally: Haruka Tsutsumi, Tomohiro Kuroda. ✉email: y-goto@chem.s.u-tokyo.ac.jp; hsuga@chem.s.u-tokyo.ac.jp

A zoles, such as oxazoles and thiazoles, are five-membered heterocycles often found in the backbone of peptidic natural products[1,2]. Such azole-containing natural peptides exhibit a variety of bioactivities, including antitumor, antifungal, antibiotic, and antiviral activities[3–9]. Incorporation of azoles into the polyamide backbone generally enhances structural rigidity, which can create conformationally preorganized scaffolds[10,11], thereby resulting in potent binding to the targets as well as better membrane permeability[12]. The local azole moieties themselves often act as key building blocks in the bioactivities, as they can interact with nucleic acids[13], proteins[14], and metals[15]. Thus, azoles are attractive entities to have in not only natural products but also artificial peptidomimetics[16–19] in devising therapeutically valuable molecules.

Since the translation apparatus allows for the facile synthesis of various peptides in an mRNA template-dependent manner, our group and others have engaged in the use of in vitro translation systems to express nonstandard peptides containing diverse and exotic building blocks, such as β/γ-amino acids[20–23], amino carbothioic acids[24], and even non-amino acids[25–29]. However, installation of an azole ring(s) into the peptide backbone by translation remains a major challenge. To the best of our knowledge, the only adequate approach reported in the literature is the use of a mutated ribosome that incorporates azole-containing dipeptide units into the nascent peptide chain[30–34]. Unfortunately, the incorporation of such azole-containing dipeptide units into the nascent peptide chain was very modest where only 1–7% efficiency was observed for the expression of green fluorescent protein (GFP) mutants compared to the wild-type GFP expression even using the dedicated mutant ribosome (i.e., the conventional wildtype ribosome could not do the job at all)[31,32]. This implies that direct incorporation of azole moiety via such a peptidyl-transfer reaction is yet an arduous task.

Posttranslational chemical modification of peptides could offer an alternative approach to generate azole-containing peptides. Several biomimetic conversions of Cys/Ser/Thr analogs into azole moieties have been reported[35–41], where the 2-step enzymatic conversion (ATP-mediated cyclodehydration[42–44] followed by FMN-dependent oxidation[45,46]) found in natural product biosynthesis pathways[47,48] is mimicked by appropriate chemical reagents. But, the requirement of water-sensitive reagents under rather harsh conditions would prohibit achieving our goal for the posttranslational modification on peptides in aqueous media. In another method, α-chloroglycine esters could be chemically modified to β,γ-alkynylglycine esters, undergoing base-induced cyclization to yield oxazoles in aqueous media[49]. However, as such amino acid monomers are rather labile in water, the preparation of the corresponding aminoacyl-tRNA would be difficult to achieve and thereby inapplicable to ribosomal synthesis. Altogether, combinations of contemporary modification reactions and artificial amino acids currently known in the literature seemed inapplicable to the translation system.

Here, we design a translation-compatible artificial amino acid based on a 4-bromovinylglycine derivative (BrvG). Incorporation of this amino acid into the nascent peptide chain enables us to execute the posttranslational installation of azoles (oxazole and thiazole) at designated positions. Thus, this method allows for the synthesis of designer azole-containing peptides in an mRNA-dependent manner under the reprogrammed genetic code.

## Results

**Design and in vitro ribosomal incorporation of BrvG.** To achieve chemical posttranslational modification yielding azoles, we have conceived the ribosomal incorporation of BrvG via genetic code reprogramming powered by flexizymes and a

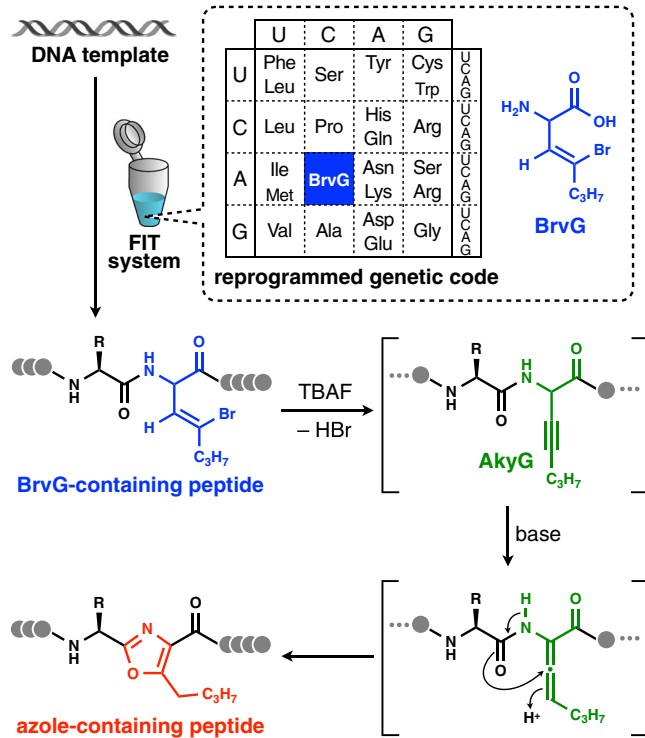

**Fig. 1 Schematic illustration of the chemical posttranslational modification for in vitro ribosomal synthesis of azole-containing peptides.** For posttranslational installation of azoles into translated peptides, a 4-bromovinylglycine derivative (BrvG, shown in blue) is incorporated into the nascent peptide chain via genetic code reprogramming. Subsequent treatment of the expressed peptide with tetrabutylammonium fluoride (TBAF) induces dehydrobromination to give the corresponding β,γ-alkynylglycine derivative (AkyG, shown in green), which spontaneously undergoes isomerization to produce an azole ring (shown in red) in the peptide backbone.

custom-made cell-free translation, so-called flexible in vitro translation (FIT) system[50] (Fig. 1). The expressed BrvG-containing peptide can be subjected to tetrabutylammonium fluoride (TBAF)-induced dehydrobromination[51] conditions, where the BrvG residue can be specifically converted to the corresponding β,γ-alkynylglycine derivative (AkyG). It has been reported that the β,γ-alkynyl group is prone to react with an adjacent amide carbonyl group under mild basic conditions via an allene intermediate[49,52] to yield a backbone oxazole moiety. Thus, we have envisioned that the AkyG-containing peptide generated in situ can be spontaneously converted to the corresponding peptide bearing oxazole.

To utilize BrvG in reprogrammed genetic codes, BrvG bearing 3,5-dinitrobenzyl ester (BrvG-DBE) was chemically synthesized (Supplementary Fig. 1). To confirm that BrvG can be charged onto tRNA by means of the flexizyme technology, we used our conventional assay system using a tRNA analog, microhelix RNA (μhRNA)[53]. The BrvG-DBE substrate was incubated with μhRNA in the presence of a flexizyme (dFx), and the resulting products were analyzed by acidic polyacrylamide gel electrophoresis (PAGE) to evaluate the aminoacylation efficiency (Supplementary Fig. 2a). The gel showed a mobility-shifted band corresponding to the BrvG-μhRNA, indicating that the flexizyme system allowed for the preparation of BrvG-tRNA under these conditions. To ribosomally synthesize a model peptide bearing a BrvG residue (Pep1-Br), BrvG-tRNA$^{AsnE2}_{GGU}$ was added to a Thr-depleted FIT system, in which the ACC codon was reprogrammed with BrvG (Supplementary Fig. 2b). Tricine-SDS PAGE analysis of the

translation product revealed that BrvG was efficiently (86% compared with the wildtype expression) incorporated into the nascent peptide chain (Supplementary Fig. 2c). In addition, matrix-assisted laser desorption/ionization time-of-flight mass spectrometry (MALDI-TOF-MS) of the translation product showed a peak corresponding to the expected peptide (Supplementary Fig. 2d), confirming that BrvG could be successfully utilized in the FIT system to yield Pep1-Br.

**Posttranslational chemical modification of BrvG to an oxazole.** The expressed Pep1-Br peptide was then subjected to the chemical posttranslational modification to determine whether the designed reaction could occur. Pep1-Br was incubated with TBAF in aqueous N,N-dimethylformamide (DMF) at 60 °C for 120 min (Fig. 2a). The reaction product was recovered by acetone precipitation and analyzed by MALDI-TOF-MS. The mass spectrum showed a sole peak consistent with the expected molecular mass

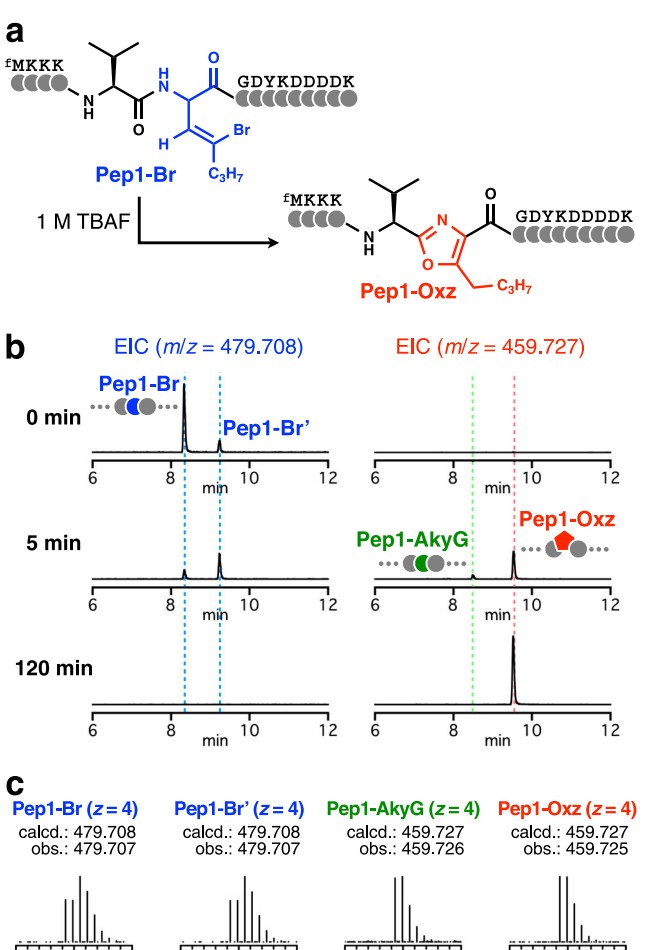

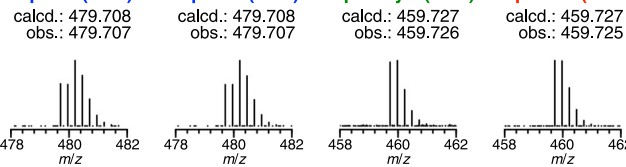

**Fig. 2 Chemical posttranslational modification of a model peptide bearing BrvG to yield an oxazole-containing peptide. a** Sequence of the model peptide Pep1-Br and its corresponding oxazole-containing peptide Pep1-Oxz. **b** LC-ESI-MS analysis to monitor the chemical posttranslational modification of Pep1-Br. The translated Pep1-Br was incubated with 1 M TBAF for 0, 5, and 120 min and analyzed by LC-ESI-MS. EICs with m/z values corresponding to the precursor (Pep1-Br/Pep1-Br', 479.708) and the expected dehydrobrominated product (Pep1-AkyG/Pep1-Oxz, 459.727) are shown. The peaks corresponding to Pep1-Br/Pep1-Br', Pep1-AkyG, and Pep1-Oxz are labeled with blue, green, and red dotted lines, respectively. Additional EICs of more time points are shown in Supplementary Fig. 3. **c** ESI mass spectra of Pep1-Br, Pep1-Br', Pep1-AkyG, and Pep1-Oxz.

of the dehydrobrominated product (Supplementary Fig. 2e). This result suggested that BrvG could undergo dehydrobromination by TBAF and that no significant side reactions on other proteinogenic amino acids occurred.

As the molecular weight of the desired oxazole-containing peptide (Pep1-Oxz) is identical to that of the expected intermediate (Pep1-AkyG), the MALDI-TOF-MS experiment above could not determine whether the oxazole moiety was generated as designed. Thus, to monitor the production of the intermediate and the desired oxazole-containing product at a finer temporal resolution, the reaction was quenched after 0, 5, 15, 30, and 120 min and analyzed by liquid chromatography-electrospray ionization-mass spectrometry (LC-ESI-MS, Fig. 2b and Supplementary Fig. 3). Before the reaction, the extracted ion chromatogram (EIC) with the m/z value corresponding to the BrvG-containing peptide showed two peaks (Pep1-Br and Pep1-Br'). Both peptides showed the same isotopic pattern, indicating the presence of a bromine atom (Fig. 2c), and had almost identical fragmentations in tandem mass spectrometry (MS/MS) (Supplementary Fig. 4), suggesting that the FIT system yielded two isomers. Given that the BrvG-DBE synthesized was a mixture of L- and D-isomers, they were likely epimers bearing L-BrvG and D-BrvG. Nonetheless, both BrvG-containing species were fully consumed within 120 min incubation. Meanwhile, EICs for the dehydrobrominated peptides did not show any peak at 0 min but gave two peaks (retention time (Rt) = 8.48 and 9.53 min) at 5 and 15 min incubation time points, and either of these peaks could correspond to Pep1-AkyG or Pep1-Oxz. The isotopic patterns indicated that no bromine atom was present in these peptides (Fig. 2c). After prolonged incubation, the two peaks converged into the latter peak (Rt = 9.53 min).

To elucidate the identity of the dehydrobrominated products, each peptide was analyzed by MS/MS. The minor peak (Rt = 8.48 min) gave fragments consistent with the expected intermediate Pep1-AkyG, in which fragmentation in almost every amide bond, including the AkyG-adjacent positions ($b_5$, $b_6$, $y_9$, and $y_{10}$ ions), was observed (Fig. 3a). On the other hand, the MS/MS spectrum of the major peak (Rt = 9.53 min) lacked the characteristic $b_5$, $b_6$, $y_9$, and $y_{10}$ ions, indicating no fragmentation at the upstream/downstream positions of the dehydrobrominated residue (Fig. 3b). This observation was consistent with previous reports that MS/MS fragmentations at the positions adjacent to backbone azoles were strongly suppressed in azole-containing peptides[54–56], suggesting that this peptide is most likely the desired oxazole-containing peptide (Pep1-Oxz). Collectively, we concluded that BrvG ribosomally incorporated into the nascent peptide chain was first converted to AkyG (Rt = 8.48 min), which further underwent isomerization yielding Oxz (Rt = 9.53 min) under the chemical posttranslational modification conditions.

**Production of oxazole-containing peptides with various sequence compositions.** To demonstrate that our method can tolerate various adjacent amino acids, we expressed Pep2(X)-Br and Pep3(X)-Br peptides, which have various arbitrarily chosen residues (X) at the upstream and downstream positions of BrvG (Supplementary Fig. 5a, b), and treated them with TBAF. In all of the 12 tested sequences, the BrvG-containing peptides were smoothly converted into two dehydrobrominated peptides after 5 min incubation, and the peptide with the smaller Rt (AkyG-containing intermediates) was further converted into the other product (Oxz-containing peptides) after 120 min incubation (Supplementary Fig. 5c, d). This result shows that the chemical posttranslational modification is compatible with a variety of proteinogenic amino acids located near the azole. This wide substrate scope illustrates the potential for the production

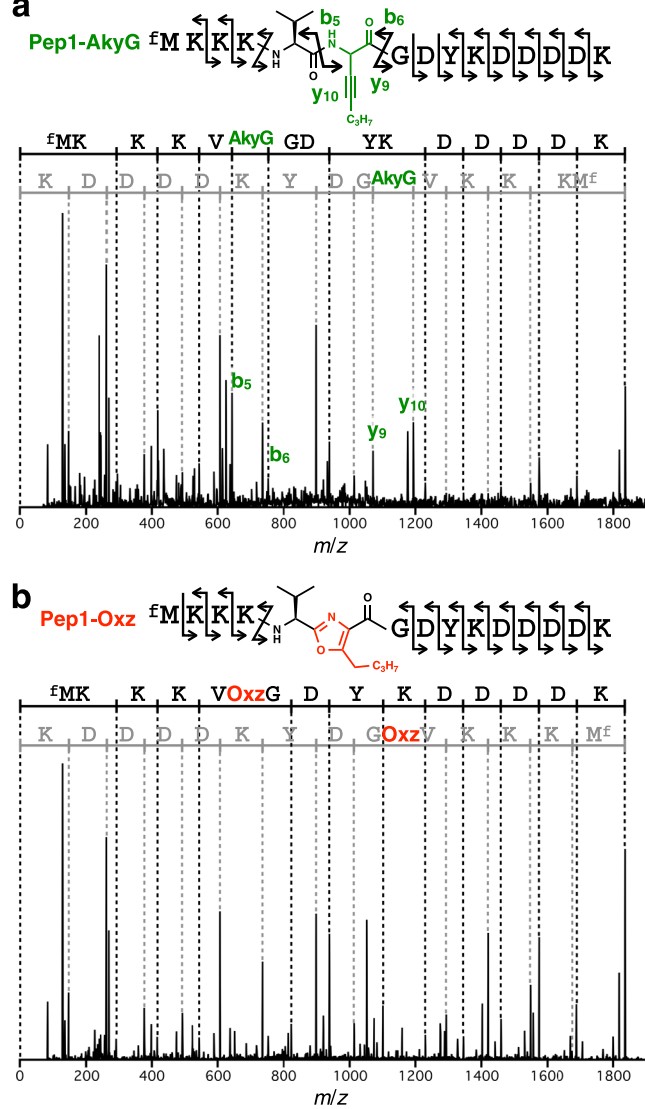

**Fig. 3 Identification of Pep1-AkyG and Pep1-Oxz by MS/MS. a, b** MS/MS spectra of peaks observed at **a** Rt = 8.48 min (Pep1-AkyG) and **b** Rt = 9.53 min (Pep1-Oxz). Observed b-ion series and y-ion series fragmentations are shown on the sequences in black and gray, respectively. The peaks corresponding to the key fragmentations that discriminate the two compounds ($b_5$, $b_6$, $y_9$, and $y_{10}$) are labeled (green).

of designer azole-containing peptides with various sequence compositions.

Although Hecht et al. elegantly demonstrated that a mutated ribosome was able to incorporate azole-containing dipeptide units into the nascent peptide chain[30–34], in principle, this methodology could not yield peptides bearing successive oxazoles. In addition, azole-forming posttranslational modification enzymes generally exhibit substrate preferences to produce certain azole patterns (e.g., azoles in successive[57], alternate[58], or every third[59] position) found in their cognate natural products. To demonstrate that the chemical posttranslational modification enables us to yield peptides having multiple oxazoles with various patterns, we designed precursor peptides bearing two BrvG residues separated by spacers (Fig. 4a, Pep4-Br-X-Br, where X indicates the spacer sequence) and two consecutive BrvG residues (Fig. 4b, Pep5-Br-Br). Upon incubation with TBAF, Pep4-Br-GV-Br and Pep4-Br-V-Br were both converted to the corresponding peptides bearing two oxazoles within 240 min (Fig. 4c,

Supplementary Fig. 6a, b). Similarly, Pep5-Br-Br was completely consumed within 5 min, but the subsequent isomerization step to generate the tandem oxazoles turned out to be relatively slow (Fig. 4d and Supplementary Fig. 6c). This observation would likely be caused by the generation of an oxazole at the upstream position, resulting in conjugation with the adjacent carbonyl and thus lowering its reactivity with AkyG at the other position. Nonetheless, prolonged incubation (960 min) led to complete conversion of Pep5-Br-Br to Pep5-Oxz-Oxz bearing tandem oxazoles. Taken together, these experiments have clearly demonstrated that our methodology is applicable to the ribosomal synthesis of various peptides containing multiple and even tandem backbone oxazoles.

**Production of oxazole-containing macrocyclic peptides**. It has been proven that the thioether-closed macrocycle offers a remarkable peptide scaffold for the discovery of peptide ligands against target proteins of interest[60]. We then wondered if our method is compatible with the ribosomal synthesis of thioether-cyclic peptides via spontaneous posttranslational cyclization between a chloroacetylated non-proteinogenic amino acid and Cys[61]. To verify this possibility, we first expressed a linear precursor peptide (linPep6-Br) containing N-terminal N-chloroacetyl-tyrosine ($^{ClAc}$Y), internal BrvG, and C-terminal Cys by means of a Met/Thr-depleted FIT system (Fig. 5a). Though the Cys thiol group in the expressed linear precursor could have potentially reacted with the BrvG moiety, such potential byproducts were not detected in LC-ESI-MS (Supplementary Fig. 7). Rather, LC-ESI-MS analysis of the translation product showed the formation of the desirable thioether-closed macrocyclic peptide bearing the intact BrvG (cyPep6-Br), indicating that the Cys thiol selectively reacted with the N-terminal ClAc group even in the presence of intramolecular BrvG. Incubation of cyPep6-Br with TBAF cleanly yielded the expected macrocyclic peptide containing an internal oxazole (cyPep6-Oxz). This result has demonstrated that the chemical posttranslational modification method allows for a ribosomal synthesis of oxazole-containing macrocyclic peptides.

**Production of thiazole-containing peptides**. During the posttranslational modification, the amide carbonyl oxygen attacks the downstream AkyG side chain and is eventually incorporated into the resulting azole; thus, in principle, this method can produce only oxazole rings on standard peptides. We previously reported that thionated alanine (Ala$^S$) could be accepted as a monomer in the FIT-mediated genetic code reprogramming, allowing for the ribosomal synthesis of thioamide-containing peptides[24]. Therefore, we envisaged that incorporation of a thioamide prior to the BrvG residue could yield peptides with a thiazole, which is broadly found in peptidic natural products. To test this strategy, a precursor peptide with an Ala$^S$-BrvG motif (Pep7-A$^S$-Br) was expressed in a His/Thr-depleted FIT system and then treated with TBAF (Fig. 5b and Supplementary Fig. 8). Compared with the above-discussed oxoamide analogs, Pep7-A$^S$-Br showed slower dehydrobromination and required longer incubation (up to 960 min) for complete consumption. Nevertheless, the generated Pep7-A$^S$-AkyG intermediate smoothly reacted and yielded the expected thiazole-containing peptide. This result has clearly demonstrated that thioamides can participate in the nucleophilic addition to the neighboring AkyG and result in thiazole, expanding the repertoire of azole structures accessible by this method.

In summary, we have demonstrated a strategy for the posttranslational incorporation of azoles into ribosomally synthesized peptides. In this methodology, the non-proteinogenic BrvG

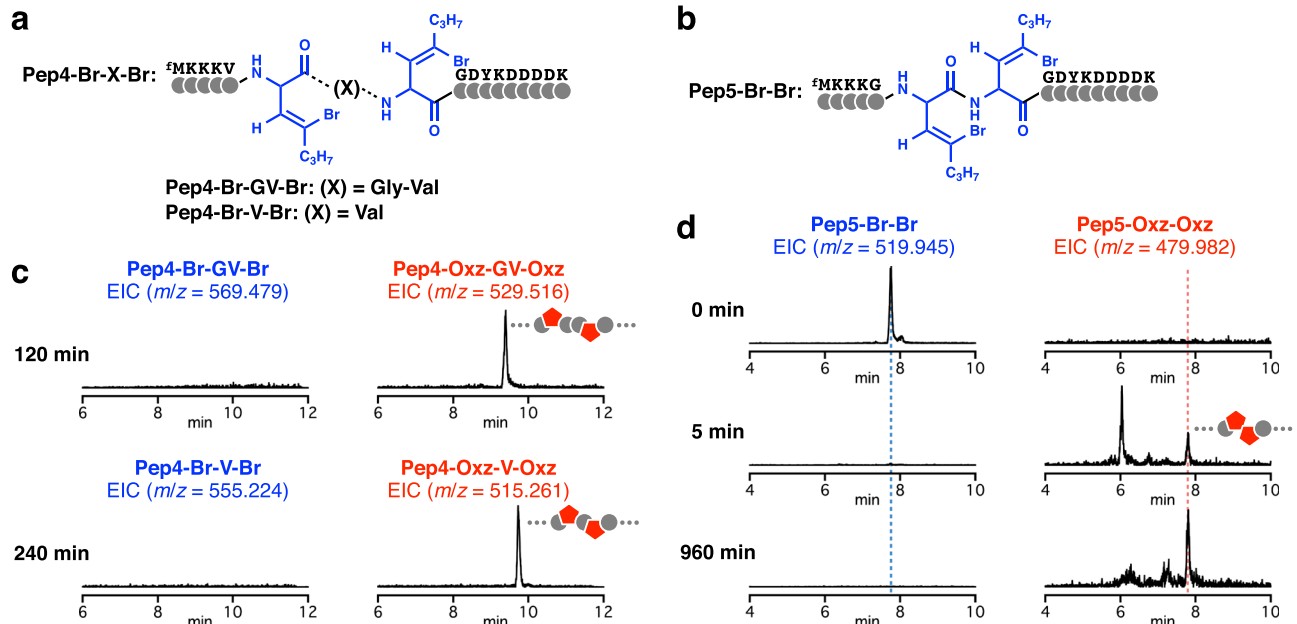

**Fig. 4 Synthesis of peptides containing multiple oxazoles. a** Sequences of Pep4-Br-X-Br containing two BrvG residues separated by different spacer regions (X). **b** Sequence of Pep5-Br-Br containing two successive BrvG residues. **c, d** Synthesis of peptides containing two oxazole moieties. EICs of *m/z* values corresponding to the precursors and the expected products with two Oxz are shown. Additional EICs used to monitor these reactions are shown in Supplementary Fig. 6.

residue incorporated via genetic code reprogramming has been utilized as a chemical handle for selective posttranslational modification to chemically generate 5-alkyl-azoles on ribosomally synthesized peptides. In contrast to enzymatic modification or biomimetic chemical conversion of Cys/Ser/Thr to yield azoles, the wide substrate scope, chemoselectivity, and aqueous reaction conditions of our methodology have enabled the mRNA-templated synthesis of designer azole-containing peptides. In principle, by modifying the γ-substitution of the 4-bromovinylglycine derivative (propyl group in BrvG), this posttranslational chemical modification reaction is potentially applicable to the synthesis of peptides bearing more diverse azole structures, though it could be limited to those with substitutions at the 5th position. More importantly, this method can be integrated with other previously devised chemical posttranslational modification reactions[62–67] to yield peptides with various non-canonical backbone structures, expanding the chemical diversity of peptidomimetics accessible by in vitro translation.

## Methods

**Ribosomal synthesis of BrvG-containing peptides**. The reconstituted in vitro translation system without 20 standard amino acids was prepared as previously described[50] by mixing purified ribosome, enzymes, and translation factors derived from *Escherichia coli*. In brief, the translation solution contained 50 mM HEPES-KOH (pH 7.6), 100 mM KOAc, 2 mM guanosine triphosphate (GTP), 2 mM adenosine triphosphate (ATP), 1 mM cytidine triphosphate (CTP), 1 mM uridine triphosphate (UTP), 20 mM creatine phosphate, 12 mM Mg(OAc)₂, 2 mM spermidine, 2 mM dithiothreitol (DTT), 1.5 mg/mL *E. coli* total tRNA (Roche), 1.2 μM ribosome, 0.6 μM methionyl-tRNA formyltransferase (MTF), 2.7 μM prokaryotic initiation factor-1 (IF1), 0.4 μM prokaryotic initiation factor-2 (IF2), 1.5 μM prokaryotic initiation factor-3 (IF3), 10 μM elongation factor thermo unstable (EF-Tu), 10 μM elongation factor thermo stable (EF-Ts), 0.26 μM elongation factor G (EF-G), 0.25 μM release factor-2 (RF2), 0.17 μM release factor-3 (RF3), 0.5 μM ribosome recycling factor (RRF), 4 μg/mL creatine kinase, 3 μg/mL myokinase, 0.1 μM pyrophosphatase, 0.1 μM nucleotide-diphosphatase kinase, 0.1 μM T7 RNA polymerase, 0.73 μM alanyl-tRNA synthetase (AlaRS), 0.03 μM ArgRS, 0.38 μM AsnRS, 0.13 μM AspRS, 0.02 μM CysRS, 0.06 μM GlnRS, 0.23 μM GluRS, 0.09 μM GlyRS, 0.02 μM HisRS, 0.4 μM IleRS, 0.04 μM LeuRS, 0.11 μM LysRS, 0.03 μM MetRS, 0.68 μM PheRS, 0.16 μM ProRS, 0.04 μM SerRS, 0.09 μM ThrRS, 0.03 μM TrpRS, 0.02 μM TyrRS, 0.02 μM ValRS, and 100 μM 10-formyltetrahydrofolate (10-HCO-H4 folate). Reaction mixtures were prepared under the conditions summarized in Supplementary Table 3.

**Tricine-SDS PAGE analysis of Pep1-Br**. For the radioisotope labeling, the translation reaction (2.5 μL) was performed in the presence of 50 μM [¹⁴C]-Asp (Perkin Elmer). The translation solution was mixed with 2.5 μL of loading buffer (900 mM Tris-HCl pH 8.45, 30% glycerol, 8% SDS) and heated at 95 °C for 5 min. The mixture was applied to tricine-SDS PAGE gel (15% acrylamide, 1 M Tris-HCl pH 8.45, 10% glycerol, 0.1% SDS) and electrophoresis was carried out at 150 V for 30 min. The gel was dried, imaged by autoradiography using a Typhoon FLA 7000 (GE Healthcare) under control of FLA 7000 software v.1.2, and analyzed by ImageQuant TL v.8.1 (GE Healthcare). The concentrations of the expressed peptides were determined by comparison with serially diluted [¹⁴C]-Asp with known concentrations as standards.

**Posttranslational chemical modification of BrvG-containing peptides**. The translation mixture (2.5 μL) was mixed with 7.5 μL of acetone and cooled to −20 °C. The resulting mixture was incubated at −20 °C for more than 30 min and centrifuged at 4 °C, 15,300 × g for 5 min. The supernatant was removed and the pellet was air-dried for 5 min. The pellet was resuspended in 1 μL of water and mixed with 1.5 μL of DMF and 22.5 μL of 1.1 M TBAF in DMF. The reaction mixture was incubated at 60 °C for various reaction times. The reaction was stopped by the addition of 50 μL of 5% TFA aq. and 750 μL of acetone. The resulting mixture was incubated at −20 °C for more than 30 min and centrifuged at 4 °C, 15,300 × g for 5 min. The supernatant was removed and the pellet was air-dried for 5 min to collect the product of chemical posttranslational modification.

**MALDI-TOF mass spectrometry**. The pellet of the product of chemical posttranslational modification was resuspended in 5.0 μL of 1×TBS (50 mM Tris-HCl pH 7.6 and 150 mM NaCl), and full-length peptides were pulled down by Anti-Flag M2 affinity gel (Sigma) for 1 h. The peptides were eluted with 5.0 μL of 0.2% TFA, loaded on a solid-phase extraction (SPE) column (C-Tip C18; Nikkyo Technos) and eluted with a matrix solution (80% acetonitrile, 0.5% acetic acid, and half-saturated α-cyano-4-hydroxycinnamic acid (Bruker Daltonics)). MALDI-TOF-MS was carried out on an ultrafleXtreme (Bruker Daltonics) under control of flex-Control v.3.4 in reflector mode, externally calibrated with peptide calibration standard II (Bruker Daltonics). The obtained mass spectra were analyzed by flexAnalysis v.3.4.

**LC-ESI-qTOF mass spectrometry**. The pellet of the product of chemical posttranslational modification was resuspended in 30 μL of 5% TFA aq. An aliquot (10 μL) of the resulting solution was separated on an Acquity UPLC (Waters) equipped with a C18 column (Peptide BEH C18 Column, 300 Å, 1.7 μm, 2.1 mm × 150 mm, Waters) using a gradient of 5–40% B over 10 min (A = water containing 0.1% formic acid, B = acetonitrile containing 0.1% formic acid), and directly subjected to an ESI-qTOF mass spectrometer (Waters Xevo G2-XS system, under control of MassLynx v.4.1). Nitrogen was used as cone gas (50 L/min) and desolvation gas (1000 L/min). The capillary voltage was set to 0.7 kV. The ionization source and

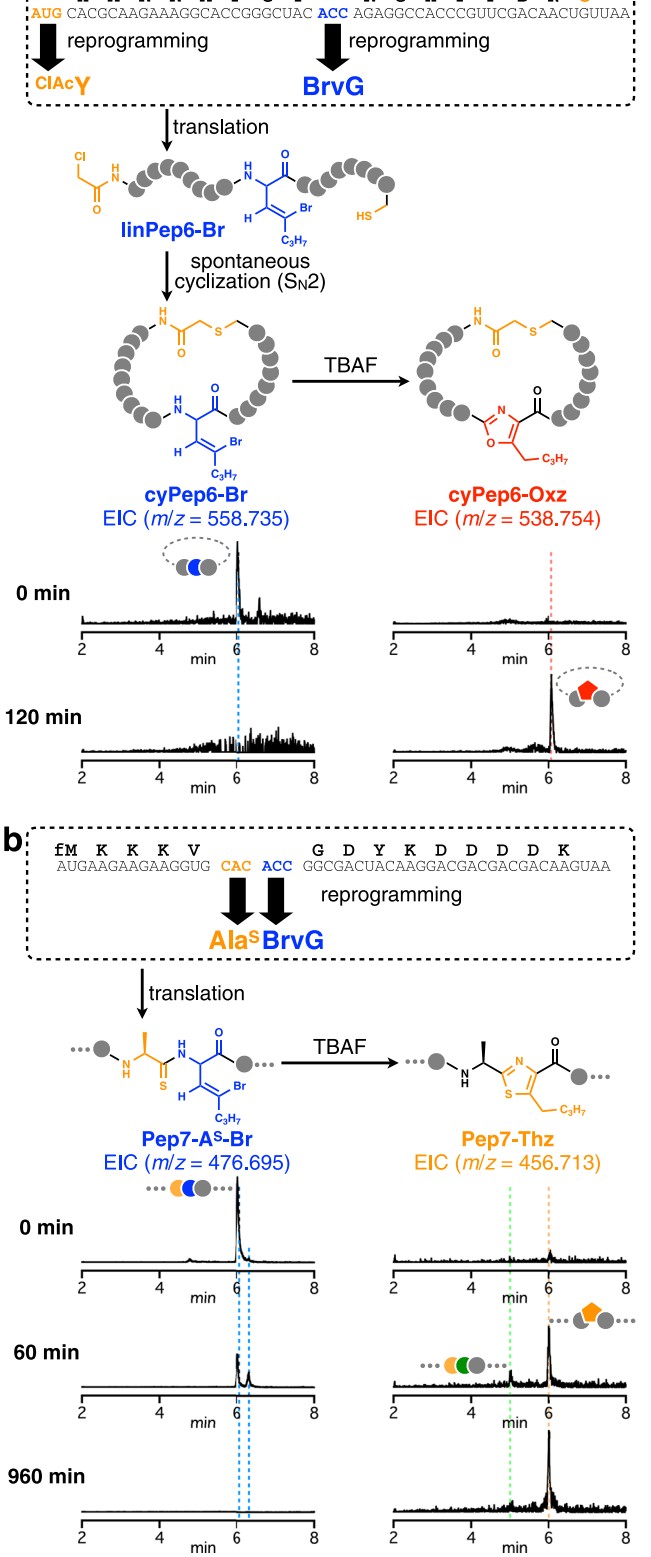

**a**
```
    H A R K A P G Y    R G H P F D N  C
AUG CACGCAAGAAAGGCACCGGGCUAC ACC AGAGGCCACCCGUUCGACAACUGUUAA
```
↓ reprogramming    ↓ reprogramming
ClAcY               BrvG

↓ translation

**linPep6-Br**

spontaneous cyclization ($S_N2$)

**cyPep6-Br**
EIC ($m/z$ = 558.735)

→ TBAF →

**cyPep6-Oxz**
EIC ($m/z$ = 538.754)

0 min

120 min

**b**
```
fM  K K K V       G D Y K D D D D K
AUGAAGAAGAAGGUG CAC ACC GGCGACUACAAGGACGACGACGACAAGUAA
```
reprogramming
Ala$^S$ BrvG

↓ translation

**Pep7-A$^S$-Br**
EIC ($m/z$ = 476.695)

→ TBAF →

**Pep7-Thz**
EIC ($m/z$ = 456.713)

0 min

60 min

960 min

**Fig. 5 Application of the chemical posttranslational modification of BrvG for the synthesis of various peptides bearing azole moieties. a** Ribosomal synthesis of a thioether-closed macrocyclic peptide bearing a backbone oxazole. **b** Ribosomal synthesis of a peptide bearing a thiazole moiety. The peaks corresponding to translated precursors, intermediates bearing AkyG, the cyclic peptide with an oxazole, and the thiazole-containing peptide are labeled with blue, green, red, and orange dotted lines, respectively. Additional EICs of more time points monitored during the synthesis of Pep7-Thz are shown in Supplementary Fig. 8.

**Reporting summary**. Further information on research design is available in the Nature Research Reporting Summary linked to this article.

## Data availability
Primer sequences, DNA template assembly schemes, characterization of BrvG-DBE, gel electrophoresis, and MALDI-TOF-MS data to demonstrate genetic code reprogramming using BrvG, MS/MS identification of Pep1-Br, and summary of all LC–MS data discussed in this paper are available in Supplementary Information. Other results are available from the corresponding authors upon reasonable request.

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

## Acknowledgements

This work was partially supported by KAKENHI (JP16H06444 to H.S. and Y.G.; JP20H05618 to H.S.; JP15K12739, JP17H04762, JP18H04382, JP19H01014, JP19K22243, JP20H02866 to Y.G.) from the Japan Society for the Promotion of Science.

## Author contributions

Y.G. and H.S. conceived and supervised the study. All authors designed experiments. H.T. synthesized BrvG-DBE. H.T. and T.K. performed expression and modification of BrvG-containing peptides. H.K. performed quantification of the FIT-expressed BrvG-containing peptide. All authors analyzed the experimental results. H.T., Y.G., and H.S. wrote the manuscript with input from all authors. Y.G. prepared manuscript figures.

## Competing interests

The authors declare no competing interests.

**Additional information**

