## [Peer Review File · Nature Communications]

Reviewers' Comments:

Reviewer #1:

Remarks to the Author:

In this manuscript, Tsutsumi et al. present a chemical approach that enables site-specific installation of azole group into a peptide. They design an artificial amino acid bearing a unique chemical functional moiety converted to an azole group by a post-translational oxidation reaction. The authors show the bromovinylglycine (BrvG) amino acid is chargeable to synthetic tRNA by the flexizyme system and subsequently incorporated into a peptide using an in vitro protein translation platform, the so-called FIT system. This is a clever approach. They provide compelling characterization data sets obtained by chromatographic and mass spectrometric analyses (EIC and MALDI), which reveal a short-lived alkyne intermediate of the azole formation reaction. Significant in this work is that the chemical modification reaction they designed is easy, fast, efficient, and allows consecutive azole formation, which is impossible by the enzymatic reactions they have published (Chem. Lett. 2016, 45, 1247–1249, Chemistry & Biology 2014, 21, 766–774). They also show BrvG undergoes the oxidation reaction with the carbonyl and thionyl of the downstream amino acid, resulting in the formation of oxazoles and thiazoles, respectively. From an application perspective, I believe this work shows the possibility of producing various peptidomimetic drugs available through biomanufacturing. I really like this paper, but have a few concerns to be addressed:

Major Concerns:

- This work connects well to the previous studies in the genetic code expansion community. It indeed expands the scope of non-canonical substrates that can be incorporated into a peptide in vitro ribosomal synthesis. This chemistry can be a powerful tool to generate a wide variety of azole groups, however, the authors only showed the chemistry with one example of BrvG derivative. It would be more significant to investigate other BrvG variants that react and form an azole group rather than examine the compatibility with other natural amino acids located at the n-1 or n+1 position of a peptide. I feel the authors should show more BrvG derivatives are compatible with this chemical reaction.

- Along with the context above, the authors should explain why they chose BrvG with a propyl group (C₃H₇), as this thiazole derivative is rarely, if ever, observed in nature (I think). Additionally, the authors should note that their method is limited to incorporating 5-alkyl-azoles, while many azoles in natural products lack substituents at the 5 position.

- Page 6. The authors compare the substrate scope of their method to enzymatic means. First, I suggest including a short summary of the analogous enzymatic transformations in the intro with the other chemical posttranslational methods. In addition, I encourage the authors to revise the following sentences "The wide substrate scope is in stark contrast to the enzymatic posttranslational modification, which is often severely affected by types of residues adjacent to the modification site." And "In addition, azole-forming posttranslational modification enzymes generally exhibit substrate preferences to produce specific azole patterns (eg azoles in successive, alternate, or every third position) found in their cognate natural products, often limiting their potential as versatile biocatalysts to obtain azole-containing designer peptides." While adjacent residues can influence enzymatic transformation, in general, these enzymes are regarded as highly promiscuous and flexible biocatalysts. If I understand this correctly, the citations for these sentences seemingly contradict the statements made in the made text. For instance, in one of the cited works, Chem. Biol. 21, 766-774 (2014), "the FIT-PatD system unveiled unprecedented in vitro substrate tolerance of PatD and demonstrated its potential for the construction of artificial Az-peptide libraries." (Bull. Chem. Soc. Jpn. 2018, 91 (3), 410-419.) In addition, while some sequence selectivity rules were established for enzymes in the cited ACS synth. Biol. 4, 482-492 (2015), the enzymes were still able to produce 100s of unique compounds and the authors speculated that libraries of 10⁵-10⁶ modified peptides could be made. Can the authors please clarify?

Minor Concerns:

- On page 4, the authors should describe how they quench the reaction, although it is described in the methods section in SI.
- The concentration of Fx seems to be significantly higher than that of microhelix tRNA based on the gel they provided in SI. The authors should describe how much excess of Fx was used for the acylation reaction.
- Page 5/Supp Fig 4. The signal-to-noise on peptides where X = Arg or His is fairly poor compared to the other examples. The authors should note this in the text (as it seems N-term His or Arg and C-term Arg lead to decreased conversion) and comment on whether the peptides degrade or get converted to an undesired side product in these cases. Or perhaps the expression of those peptides is not as good as the others. Same comment for Figure 4 and Supp Fig 5 with multiple azoles.

Reviewer #2:

Remarks to the Author:

This manuscript by Goto, Suga and team describes the use of a post translational reaction to install azole moieties into cyclic peptide libraries produced by mRNA display. This work builds on previous work by the authors, using thionated alanine for the generation of thiamine-containing cyclic peptides.

The above is achieved by first incorporating a bromovinylglycine into a 'displayed' cyclic peptide, and triggering dehydronbromination in aqueous media. The authors go on to conclusively demonstrate that this reaction is occurring with high yield.

Overall, the paper is excellent, well written, with the data being of high quality and supporting the assertion that an azole is being formed in the assessed cyclic peptides.

My only suggestion is that while the author demonstrate the formation of azoles in a small handful of cyclic peptides, the paper would be much improved if the suitability of this method for the generation of whole mRNA display libraries containing azoles was assessed. It may be possible to do this by assessing a library containing bromovinylglycine by bromine NMR before and after the dehydronbromination reaction. But this isn't essential for this paper, and is a suggestion for a future experiment.

Overall this is an excellent paper, which will be a good fit for this journal.

Reviewer #1:

In this manuscript, Tsutsumi et al. present a chemical approach that enables site-specific installation of azole group into a peptide. They design an artificial amino acid bearing a unique chemical functional moiety converted to an azole group by a post-translational oxidation reaction. The authors show the bromovinylglycine (BrvG) amino acid is chargeable to synthetic tRNA by the flexizyme system and subsequently incorporated into a peptide using an in vitro protein translation platform, the so-called FIT system. This is a clever approach. They provide compelling characterization data sets obtained by chromatographic and mass spectrometric analyses (EIC and MALDI), which reveal a short-lived alkyne intermediate of the azole formation reaction. Significant in this work is that the chemical modification reaction they designed is easy, fast, efficient, and allows consecutive azoline formation, which is impossible by the enzymatic reactions they have published (Chem. Lett. 2016, 45, 1247–1249, Chemistry & Biology 2014, 21, 766–774). They also show BrvG undergoes the oxidation reaction with the carbonyl and thionyl of the downstream amino acid, resulting in the formation of oxazoles and thiazoles, respectively. From an application perspective, I believe this work shows the possibility of producing various peptidomimetic drugs available through biomanufacturing. I really like this paper, but have a few concerns to be addressed:

We thank this reviewer for the positive comments. According to invaluable critiques, we revised the manuscript with point-by-point responses below.

Major Concerns:

- This work connects well to the previous studies in the genetic code expansion community. It indeed expands the scope of non-canonical substrates that can be incorporated into a peptide in vitro ribosomal synthesis. This chemistry can be a powerful tool to generate a wide variety of azole groups, however, the authors only showed the chemistry with one example of BrvG derivative. It would be more significant to investigate other BrvG variants that react and form an azole group rather than examine the compatibility with other natural amino acids located at the n-1 or n+1 position of a peptide. I feel the authors should show more BrvG derivatives are compatible with this chemical reaction.

Response:

As suggested by this reviewer, there is still room to further expand the repertoire of azoles accessible via our chemical posttranslational modification by replacing the propyl group of BrvG with other substitutions. Our present work has shown an important proof-of-concept for the posttranslational chemical modification that can convert a single non-canonical amino acid (BrvG) into two different types of heterocyclic backbone, oxazole and thiazole. Although we agree that it is fascinating to diversify the azole structures by utilizing other BrvG derivatives, considerable synthetic effort is required to prepare a systematic series of BrvG derivatives with diverse types of substitutions that can competently validate such an approach. Therefore, we would like to leave such synthetic work and expansion of the repertoire to our following study.

Since we agree with this comment that suggests such a potential application of this reaction, we have added a sentence to mention this point as follows:

(Page 8, line 228ff) In principle, **by modifying the γ -substitution of the 4-bromovinylglycine derivative (propyl group in BrvG), this posttranslational chemical modification reaction is potentially applicable to the synthesis of peptides bearing more diverse azole structures, though it could be limited to those with substitutions at the 5th position.**

- Along with the context above, the authors should explain why they chose BrvG with a propyl group (C3H7), as this thiazole derivative is rarely, if ever, observed in nature (I think). Additionally, the authors should note that their method is limited to incorporating 5-alkyl-azoles, while many azoles in natural products lack substituents at the 5 position.

Response:

As discussed above, we could potentially use other BrvG derivatives with different γ -substitutions. For the synthesis of BrvG derivatives with γ -methyl and γ -ethyl substitutions we need to use propyne and 1-butyne, respectively, as the starting materials, so it would require handling of such gaseous molecules with a technically demanding task (and inconvenient). The BrvG bearing the γ -propyl group is the simplest bromovinylglycine derivative that can be derived from liquid alkyne molecules. To demonstrate the proof-of-concept, BrvG was our primary choice for this work. Indeed, as pointed out by this comment, our method has an inherent limitation: it is only applicable to azoles bearing 5-substitutions. To note this limitation, we added/revised sentences in the revised manuscript as follows.

(Page 7, line 222ff) In this methodology, the non-proteinogenic BrvG residue incorporated via genetic code reprogramming has been utilized as a chemical handle for selective posttranslational modification to chemically generate **5-alkyl**-azoles on ribosomally synthesized peptides.

(Page 8, line 228ff) In principle, by modifying the γ -substitution of the 4-bromovinylglycine derivative (propyl group in BrvG), this posttranslational chemical modification reaction is potentially applicable to the synthesis of peptides bearing more diverse azole structures, **though it could be limited to those with substitutions at the 5th position.**

- Page 6. The authors compare the substrate scope of their method to enzymatic means. First, I suggest including a short summary of the analogous enzymatic transformations in the intro with the other chemical posttranslational methods.

Response:

We appreciate this suggestion and have carefully reconsidered the contents of the introduction section. Given that this study aims at chemical installation of azoles and does not address any enzymatic modification, we wondered whether the addition of a paragraph dedicated to explaining enzymatic reactions would blur the subject of this study. Therefore, we decided to briefly mention that backbone azoles in natural peptides are generally incorporated via 2-step enzymatic modification and also added some new references related to the enzymatic transformations (42-48) in the introduction section of the revised manuscript as follows. We hope that this revision improves readability and helps the readers to smoothly understand the comparison between our chemical modification method and the analogous enzymatic transformations in the results and discussion section.

(Page 2, line 51ff) Several biomimetic conversions of Cys/Ser/Thr analogs into azole moieties have been reported³⁵⁻⁴¹, **where the 2-step enzymatic conversion (ATP-mediated cyclodehydration⁴²⁻⁴⁴ followed by FMN-dependent oxidation^{45,46}) found in natural product biosynthesis pathways^{47,48} is mimicked by appropriate chemical reagents.**

In addition, I encourage the authors to revise the following sentences "The wide substrate scope is in stark contrast to the enzymatic posttranslational modification, which is often severely affected by types of residues adjacent to the modification site." And "In addition, azole-forming posttranslational modification enzymes generally exhibit substrate preferences to produce specific azole patterns (eg azoles in successive, alternate, or every third position) found in their cognate natural products, often limiting their potential as versatile biocatalysts to obtain azole-containing designer peptides." While adjacent residues can influence enzymatic transformation, in general, these enzymes are regarded as highly promiscuous and flexible biocatalysts. If I understand this correctly, the citations for these sentences seemingly contradict the statements made in the made text. For instance, in one of the cited works, Chem. Biol. 21, 766-774 (2014), "the FIT-PatD system unveiled unprecedented in vitro substrate tolerance of PatD and demonstrated its potential for the construction of artificial Az-peptide libraries." (Bull. Chem. Soc. Jpn. 2018, 91 (3), 410-419.) In addition, while some sequence selectivity rules were established for enzymes in the cited ACS synth. Biol. 4, 482-492 (2015), the enzymes were still able

to produce 100s of unique compounds and the authors speculated that libraries of 105-106 modified peptides could be made. Can the authors please clarify?

Response:

The two sentences pointed out by this comment were not intended to claim that the enzymes are too specific for the application of library construction. Indeed, we meant to claim only that the chemical modification is less affected by substrate sequence compositions as compared with enzymatic methods. It is true that some peptide-modifying enzymes (especially those found in RiPP pathways) are promiscuous and potentially applicable to the construction of unique libraries. Nonetheless, some of these enzymes could carry out the heterocyclization reaction with some degrees of promiscuity, but depending on the enzyme it often exhibits a biased preference toward a certain local sequence composition. This property could have suppressed desired modifications of randomized peptide sequences in the application for library construction. For instance, such substrate preferences of azole-forming enzymes to adjacent residues (Ref# 50-53 in the original ms) and bis-azole patterns (Ref# 54-56 in the original ms (Ref #57-59 in the revised ms)) were described in the cited papers. Therefore, we believe that the description in these sentences is scientifically correct. Nonetheless, we do not intend to unnecessarily devalue the enzymatic methods for sure. To avoid unnecessarily contentious claims and describe the advantages more clearly, these sentences were reworded. The relevant texts now read:

(Page 6, line 157ff) The wide substrate scope **illustrates the potential for production of designer azole-containing peptides with various sequence compositions.**

(Page 6, line 161ff) In addition, azole-forming posttranslational modification enzymes generally exhibit substrate preferences to produce **certain** azole patterns (e.g., azoles in successive⁵⁷, alternate⁵⁸, or every third position⁵⁹) found in their cognate natural products. **[The clause “; often limiting their potential as versatile biocatalysts to obtain azole-containing designer peptides” in the original sentence has been removed.]**

Minor Concerns:

- On page 4, the authors should describe how they quench the reaction, although it is described in the methods section in SI.

Response:

In the revised manuscript, we specified how the reaction was quenched in the main text as follows.

(Page 4, line 107ff) Pep1-Br was incubated with TBAF in aqueous *N,N*-dimethylformamide (DMF) at 60°C for 120 min (Fig. 2a). **The reaction product was recovered by acetone precipitation** and analyzed by MALDI-TOF-MS.

- The concentration of Fx seems to be significantly higher than that of microhelix tRNA based on the gel they provided in SI. The authors should describe how much excess of Fx was used for the acylation reaction.

Response:

As described in the methods section (supplementary information), the concentration of dFx used is identical to that of RNA substrates (μ hRNA and tRNAs). We speculate that μ hRNA would be less efficiently stained by ethidium bromide because of its short length. Such an observation that the bands corresponding to flexizymes appear more intense than μ hRNA bands is general in our experiments, as we have reported in many previous papers (e.g. Nat. Protoc., 2011, 6, 779).

- Page 5/Supp Fig 4. The signal-to-noise on peptides where X = Arg or His is fairly poor compared to the other examples. The authors should note this in the text (as it seems N-term His or Arg and C-term Arg lead to decreased conversion) and comment on whether the peptides degrade or get converted to an undesired side product in these cases. Or perhaps the expression of those peptides is not as good as the others. Same comment for Figure 4 and Supp Fig 5 with multiple azoles.

Response:

As described in the methods section (supplementary information), the LC injection volume of some sequences (Pep2(Arg)/Pep2(His)/Pep3(Arg)) in Supplementary Fig. 4 of the original manuscript was intentionally decreased to 1 μL or 5 μL , while 10 μL of sample solution was subjected to LC-MS for all other peptides. This is the potential cause for the modest signal-to-noise ratios of these chromatograms. The reason for this method modification is because 10 μL injection of these peptides somehow caused peak broadening in the chromatograms. We did not elucidate the cause of such peak broadening but managed to analyze these peptides with better peak shapes by optimizing the LC injection amount. We prepared a figure to illustrate the effect of injection amount on peak width in some representative peptides (shown below). For Pep5-Oxz-Oxz (Fig. 4 and Supplementary Fig. 5), we speculate that the modest signal-to-noise ratio may be simply caused by its relatively poorer efficiency of ESI ionization.

Most importantly, for all peptides, we carefully analyzed the LC-MS results and did not detect any significant byproducts derived from undesirable degradation/cleavage of the peptides. Nonetheless, according to this comment, we have decided to replace some data obtained by 1 μL /5 μL injection in the original manuscript with 10- μL injected results for simplicity. In the revised manuscript, the corresponding chromatographs (Pep2(Arg)/Pep2(His)/Pep3(Arg)) in Supplementary Fig. 4 show broader peaks, but significantly better signal-to-noise ratios.

Figure for review response. Comparisons of EICs of Pep2(His) and Pep2(Arg) obtained by different injection amounts.

Supplementary Fig. 4 in the revised manuscript. The data for Pep2(Arg), Pep2(His), and Pep3(Arg) were changed from those shown in the original manuscript.

Reviewer #2:

This manuscript by Goto, Suga and team describes the use of a post translational reaction to install azole moieties into cyclic peptide libraries produced by mRNA display. This work builds on previous work by the authors, using thionated alanine for the generation of thiamine-containing cyclic peptides.

The above is achieved by first incorporating a bromovinylglycine into a 'displayed' cyclic peptide, and triggering dehydronbromination in aqueous media. The authors go on to conclusively demonstrate that this reaction is occurring with high yield.

Overall, the paper is excellent, well written, with the data being of high quality and supporting the assertion that an azole is being formed in the assessed cyclic peptides.

My only suggestion is that while the author demonstrate the formation of azoles in a small handful of cyclic peptides, the paper would be much improved if the suitability of this method for the generation of whole mRNA display libraries containing azoles was assessed. It may be possible to do this by assessing a library containing bromovinylglycine by bromine NMR before and after the dehydrobromination reaction. But this isn't essential for this paper, and is a suggestion for a future experiment.

Response:

We agree with this reviewer that the demonstration of the modification of mRNA-displayed library, if it could be included in this work, will give much higher impact. Although we appreciate the suggestion that the use of bromine NMR may be potentially effective for the assessment, we are still concerned that such an experiment is technically difficult for two main reasons. First, ⁷⁹Br and ⁸¹Br nuclei are both quadrupolar resulting in broad signals, and therefore ⁷⁹Br and ⁸¹Br NMR applications for organic/biological compounds are scarcely

carried out (Ref: DOI:10.17229/jdit.2016-0618-021). Second, the sensitivities of $^{79}\text{Br}/^{81}\text{Br}$ NMR may not be high enough (relative receptivity to ^1H is approximately 4%) since the concentration of the expressed peptide in our FIT system is as low as $\sim 4\ \mu\text{M}$ as shown in Supplementary Fig. 1.

As probably this reviewer knows, the construction of an mRNA-displayed peptide library and demonstrate the selection against a protein target takes another major effort to verify the binding ability and bioactivity of selected azole-containing peptides, which may take a year or more. We believe that the significance of our finding described in this manuscript is sufficient for a report, particularly as a communication format. Such applications for mRNA-displayed peptide libraries and their in vitro selection are our clear directions and we hope that we can report in future manuscripts.

Overall this is an excellent paper, which will be a good fit for this journal.

Response:

Thank you for the warm appraisal of our work.

Additional revisions

According to the editorial request, we have added a "Data Availability" section in the revised manuscript. Also, the editorial policy checklist and the reporting summary checklist are included in the resubmission.

Reviewers' Comments:

Reviewer #1:

Remarks to the Author:

The reviewers have addressed my concerns.

Reviewer #1:

The reviewers have addressed my concerns.

We thank this reviewer for appreciating our study.